# Body fat prediction through feature extraction based on anthropometric and laboratory measurements

Zongwen Fan[1,2], Raymond Chiong[1]*, Zhongyi Hu[3], Farshid Keivanian[1], Fabian Chiong[4]

1 School of Information and Physical Sciences, The University of Newcastle, Callaghan, NSW, Australia, 2 College of Computer Science and Technology, Huaqiao University, Xiamen, China, 3 School of Information Management, Wuhan University, Wuhan, China, 4 Alice Springs Hospital, The Gap, NT, Australia

* Raymond.Chiong@newcastle.edu.au

**Data Availability Statement:** All relevant data are within the paper.

## Abstract

Obesity, associated with having excess body fat, is a critical public health problem that can cause serious diseases. Although a range of techniques for body fat estimation have been developed to assess obesity, these typically involve high-cost tests requiring special equipment. Thus, the accurate prediction of body fat percentage based on easily accessed body measurements is important for assessing obesity and its related diseases. By considering the characteristics of different features (e.g. body measurements), this study investigates the effectiveness of feature extraction for body fat prediction. It evaluates the performance of three feature extraction approaches by comparing four well-known prediction models. Experimental results based on two real-world body fat datasets show that the prediction models perform better on incorporating feature extraction for body fat prediction, in terms of the mean absolute error, standard deviation, root mean square error and robustness. These results confirm that feature extraction is an effective pre-processing step for predicting body fat. In addition, statistical analysis confirms that feature extraction significantly improves the performance of prediction methods. Moreover, the increase in the number of extracted features results in further, albeit slight, improvements to the prediction models. The findings of this study provide a baseline for future research in related areas.

## 1 Introduction

Obesity, characterised by excess body fat, is a medical problem that increases one's risk of other diseases and health issues, such as cardiovascular diseases, diabetes, musculoskeletal disorders, depression and certain cancers [1–3]. These diseases could result in escalating the spiralling economic and social costs of nations [4]. Conversely, having extremely low body fat is also a significant risk factor for infection in children and adolescents [5], and it may cause pubertal delay [6], osteoporosis [7] and surgical complications [8]. Thus, the accurate prediction of both excess and low body fat is critical to identifying possible treatments, which would prevent serious health problems. Although a huge volume of medical data is available from

**Funding:** This research was supported by the Australian Government Research Training Program through PhD scholarships awarded to ZF and FK.

**Competing interests:** The authors have declared that no competing interests exist.

sensors, electronic medical health records, smartphone applications and insurance records, analysing the data is difficult [9]. There are often too many measurements (features), leading to the curse of dimensionality [10] from a data analytics viewpoint. With a relatively small size of patient samples, but a large number of disease measurements, it is very challenging to train a highly accurate prediction model [11]. In addition, redundant, irrelevant or noise features may further hinder the prediction model's performance [12].

Feature extraction, as an important tool in data mining for data pre-processing, has been applied to reduce the number of input features by creating new, more representative combinations of features [13]. This process reduces the number of features without leading to significant information loss [14]. In this study, three widely used feature extraction methods are utilised to reduce features. Specifically, by analysing large interrelated features, Factor Analysis (FA) can be used to extract the underlying factors (latent features) [15]. It is able to identify latent factors that adequately predict a dataset of interest. Unlike FA, which assumes there is an underlying model, Principal Component Analysis (PCA) is a descriptive feature reduction method that applies an optimal set of derived features, extracted from the original features, for model training [16]. PCA data projection concerns only the variances between samples and their distribution. Independent Component Analysis (ICA), a technique that assumes the data to be the linear mixtures of non-Gaussian independent sources [17], is widely used in blind source separation applications [18].

Feature extraction has been widely used in the medical area to map redundant, relevant and irrelevant features into a smaller set of features from the original data [19, 20]. For example, Das et al. [21] applied feature extraction methods to extract significant features from the raw data before using an Artificial Neural Network (ANN) model for medical disease classification. Their results showed that feature extraction methods could increase the accuracy of diagnosis. Tran et al. [22] proposed an improved FA method for cancer subtyping and risk prediction with good results. Sudharsan and Thailambal [23] applied PCA to pre-process the experimental datasets used for predicting Alzheimer's disease. Their results showed that applying PCA for pre-processing could improve the precision of the prediction model. In the work of Franzmeier et al. [24], ICA was utilised to extract features from cross-sectional data for connectivity-based prediction of tau spreading in Alzheimer's disease with impressive results.

In addition, machine learning methods have been increasingly applied to solve body fat prediction problems [25]. Shukla and Raghuvanshi [26] showed that the ANN model is effective for estimating the body fat percentage using anthropometric data in a non-diseased group. Kupusinac et al. [27] also employed ANNs for body fat prediction and achieved high prediction accuracy. Keivanian et al. [28, 29] considered a weighted sum of body fat prediction errors and the ratio of features, and optimised the prediction using a metaheuristic search-based feature selection-Multi-Layer Perceptron (MLP) model (MLP is a type of ANN). Chiong et al. [30] proposed an improved relative-error Support Vector Machine (SVM) for body fat prediction with promising results. Fan et al. hybridised a fuzzy-weighted operation and Gaussian kernel-based machine learning models to predict the body fat percentage, while Uçar et al. [31] combined a few machine learning methods (e.g. ANN and SVM) for the same purpose, and their models achieved satisfactory predictions.

In this study, we apply FA, PCA and ICA to extract critical features from the available features, using four machine learning methods—MLP, SVM, Random Forest (RF) [32], and eXtreme Gradient Boosting (XGBoost) [33]—to predict the body fat percentage. We consider five metrics, that is, the mean absolute error (*MAE*), standard deviation (*SD*), root mean square error (*RMSE*), robustness (*MAC*) and efficiency, in the evaluation process. We use experimental results based on real-world body fat datasets to validate the effectiveness of feature extraction for body fat prediction. One of the datasets is from the StatLib, based on body

circumference measurements [34]; the other dataset is from the National Health and Nutrition Examination Survey (NHANES) based on physical examinations [35]. In addition, we employ the Wilcoxon rank-sum test [36] to validate whether the prediction accuracy based on feature extraction improves significantly or not. The motivation of this study is to assess and compare different feature extraction methods for body fat prediction as well as provide a baseline for future research in related areas. It is worth pointing out that the results presented here are new in the context of body fat prediction. We also explore the optimal number of features used for each of the feature extraction methods while balancing accuracy and efficiency.

The rest of this paper is organised as follows: Section 2 briefly introduces the feature extraction methods and prediction models. In Section 3, experimental results based on the real-world body fat datasets are provided; specifically, performance measurements are first described, and then experimental results based on feature extraction for the prediction of body fat percentage are discussed. Lastly, Section 4 concludes this study and highlights some future research directions.

## 2 Methods

In this section, we first discuss three widely used feature extraction methods: FA, PCA and ICA. Then, we present four well-known machine learning algorithms—MLP, SVM, RF and XGBoost.

### 2.1 Feature extraction methods

Feature extraction methods are widely used in data mining for data pre-processing [37]. They can reduce the number of input features without incurring much information loss [38]. In this case, they can alleviate the overfitting of prediction models by removing redundant, irrelevant or noise measurements/features. In addition, with less misleading features, the model accuracy and computation time could be further improved.

**2.1.1 Factor analysis.** This widely used statistical method for feature extraction is an exploratory data analysis method. FA can be used to reduce the number of observable features with a set of fewer latent features (factors) without losing much information [39]. Each latent feature is able to describe the relationships between the corresponding observed features. Since the factor cannot be directly measured with a single feature, it is measured through the relationships in a set of common features, if and only if one of these requirements is satisfied: (a) The minimum number of features is used to capture maximum variability in the data and (b) the information overlap among the factors is minimised. By doing so, (1) the most common variance between features is extracted by the first latent factor; (2) eliminating the factor extracted in (1), the second factor with the most variance between the remaining features is extracted; and (3) steps (1) and (2) are repeated until the rest of features are tested. FA is very helpful for reducing features in a dataset where a large number of features can be presented by a smaller number of latent features. An example of the relationship between a factor and its observed features is given in Fig 1, in which $p$ denotes the number of observed features. If the models has $k$ latent features, then the assumption in FA is given in Eq 1. Generally, FA calculates a correlation matrix based on the correlation coefficient to determine the relationship for each pair of features. Then, the factor loadings are analysed to check which features are loaded onto which factors where factor loadings can be estimated using maximum likelihood [40].

$$Feature_i = \sum_{r=1}^{k} w_{ir} Factor_r + e_i, \tag{1}$$

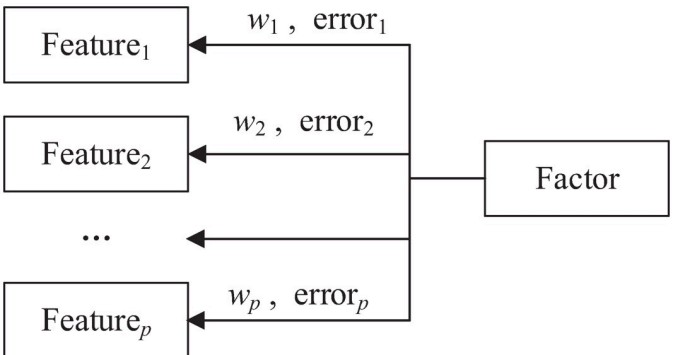

**Fig 1. An example of the relationship between a factor and its observed features.**

where $\{\{w_{ir}\}_{i=1}^{p}\}_{j=1}^{k}$ are factor loadings, which means that $w_{ir}$ is the factor loading of the $i$th variable on the $r$th factor (similar to weights or strength of the correlation between the feature and the factor) [41], and $e_i$ is the error term, which denotes the variance in each feature that is unexplained by the factor.

**2.1.2 Principle component analysis.** PCA is a very useful tool for reducing the dimensionality of a dataset, especially when the features are interrelated [42]. This non-parametric method uses an orthogonal transformation to convert a set of features into a smaller set of features termed principal components. Using a covariance matrix, we are able to measure the association of each feature with other features. To decompose the covariance matrix, singular value decomposition [43] can be applied for linear dimensionality reduction by projecting the data into a lower dimensional space, which yields eigenvectors and eigenvalues of the principal components. In this case, we could obtain the directions of data distribution and the relative importance of these directions. A positive covariance between two features indicates that the features increase or decrease together, whereas a negative covariance indicates that the features vary in opposite directions. The first principal component could preserve as much of the information in the data as possible, whereas the second one could retain as much of the remaining variability as possible until no features are left. In other words, the extracted principal components are ordered in terms of their importance (variance). Considering that PCA is sensitive to the relative scaling of the original features, in practice, it is better to normalise the data before using PCA. An example of using a component to represent its corresponding features is given in Fig 2. As this figure shows, each component is a linear function of its corresponding features, whereas a feature in FA is a function of given factors plus an error term.

**2.1.3 Independent component analysis.** ICA is a blind source separation technique [44]. It is very useful for finding factors hidden behind random signals, measurements or features based on high-order statistics. The purpose of ICA is to minimise the statistical dependence of the components of the representation. By doing so, the dependency among the extracted signals is eliminated. To achieve good performance, some assumptions should be met before using ICA [45]: (1) The source signals (features) should be statistically independent; (2) the mixture signals should be linearly independent from each other; (3) the data should be centred (zero-mean operation for every signals); and (4) the source signals should have a non-Gaussian distribution. One widely used application of ICA is the cocktail party problem [46]. As Fig 3 illustrates, there are two people speaking, and each has a voice signal. These signals are received by the microphones, which then send the mixture signals. Since the distance between the microphones and the people differ, the mixture signals from microphones differ as well.

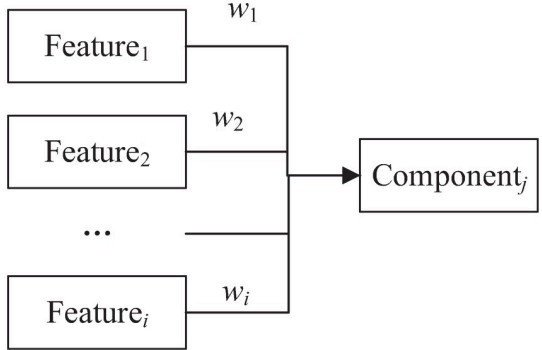

$$\text{Component}_j = w_1\text{Feature}_1 + w_2\text{Feature}_2 + \cdots + w_i\text{Feature}_i$$

**Fig 2. An example of using a component to represent its corresponding features.**

Using ICA for signal extraction, the original signals can be obtained. Notably, it is difficult for FA and PCA to extract source signals (original components).

## 2.2 Prediction models

In this section, four widely used machine learning models—MLP, SVM, RF and XGBoost—are introduced.

**2.2.1 MLP.** The MLP is a type of ANN that generally has three different kinds of layers, including the input, hidden and output layers [47]. Each layer is connected to its adjacent layers. Similarly, each neuron in the hidden and output layers is connected to all the neurons in the previous layer with a weight vector. The values from the weighted sum of inputs and bias term are fed into a non-linear activation function as outputs for the next layer. Fig 4 shows an example of MLP with three, two and one input, hidden and output neurons, respectively. We can see from the figure that the input layer has three input neurons ($x_1$, $x_2$, $x_3$) and one bias term with a value of $b^1$. Their values, based on the inner product with the weight matrix, are fed into the hidden layer. In this step, the input is first transformed using a learned non-linear transformation—an activation function $g(\cdot)$—that projects the input data into a new space where it becomes linearly separable. The outputs of two neurons in the hidden layer depend on the outputs of input neurons and a bias neuron in the same layer with a value of $b^2$. The output layer has one neuron that takes inputs from the hidden layer with the activation function, where $f(x)$ is the feed-forward prediction value from an input vector $x$.

**2.2.2 SVM.** SVMs, founded on the structural risk minimisation principle and statistical learning theory [48], have been widely used in many real-world applications and have

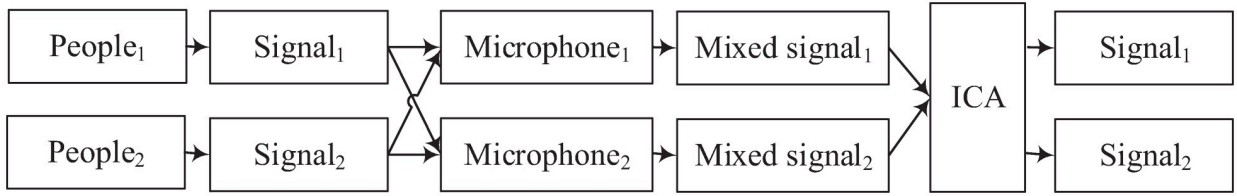

**Fig 3. An example of the process of extracting signals from the cocktail party problem with two speaking people (source signals) and two microphones (mixture signals).**

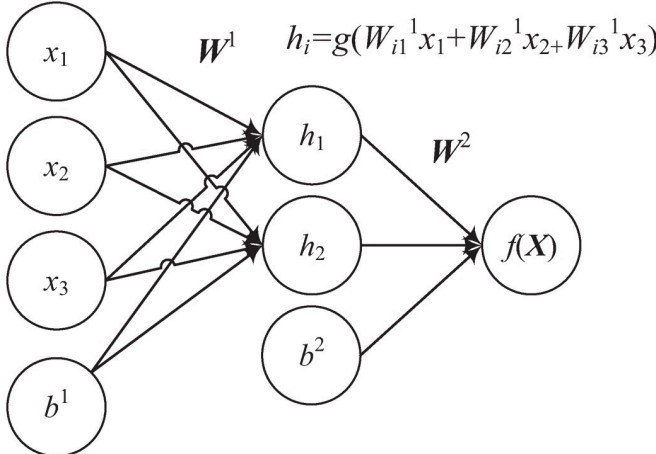

**Fig 4. An example of MLP with three input neurons, two hidden neurons, and one output neuron.**

displayed satisfactory performance (e.g., see [49–51]). Given $n$ training samples $\{(x_i, y_i)\}_{i=1}^{n}$, the standard form of $\varepsilon$-SVM regression can be expressed as Eq (2). We can see from Fig 5 that, unlike the SVM for classification problems that classifies a sample into a binary class, the SVM regression fits the best line within a threshold value $\varepsilon$ with tolerate errors ($\xi_i$ and $\xi_i^*$).

$$\arg\min_{w,b,\xi_i,\xi_i^*} \frac{1}{2}w^T w + C\sum_i (\xi_i + \xi_i^*)$$

$$s.t. \quad \begin{cases} y_i - (w^T\phi(x_i) + b) \leqslant \varepsilon + \xi_i \\ \\ (w^T\phi(x_i) + b) - y_i \leqslant \varepsilon + \xi_i^* \\ \\ \xi_i, \xi_i^* \geqslant 0 \end{cases} \tag{2}$$

where $w$ is a weight vector, $w^T$ is the transpose of $w$, $b$ is a bias term, $\xi_i$ and $\xi_i^*$ are slack variables of the $i$th sample, $C$ is a penalty parameter, $\varepsilon$ is a tolerance error, $x_i$ and $y_i$ are the $i$th input vector and output value, respectively, and $\phi(x)$ is a function that is able to map a sample from a low dimension space to a higher dimension space.

After solving the objective function in Eq (2) using the Lagrangian function [52] and Karush–Kuhn–Tucker conditions [53], we can obtain the best parameters ($\bar{w}$ and $\bar{b}$) for the SVM. The final prediction model, $g(x)$, can be expressed as follows:

$$g(x) = \sum_i (\bar{\alpha}_i - \bar{\alpha}_i^*) Kernel(x_i, x) + \bar{b}, \tag{3}$$

where $Kernel(x_i, x_j) = \phi(x_i)\phi(x_j)$ is a kernel function [54].

**2.2.3 RF.** The RF, proposed by Ho [55], is a decision tree-based ensemble model. For body fat prediction, the RF regression model uses an ensemble learning method for regression. It creates many decision trees based on the training set [56]. By combining multiple decision trees into one model, the RF model improves the prediction accuracy and stability. It is also able to avoid overfitting by utilising resampling and feature selection techniques. The training procedure of RF is given in Fig 6. As the figure illustrates, the RF generates many sub-datasets

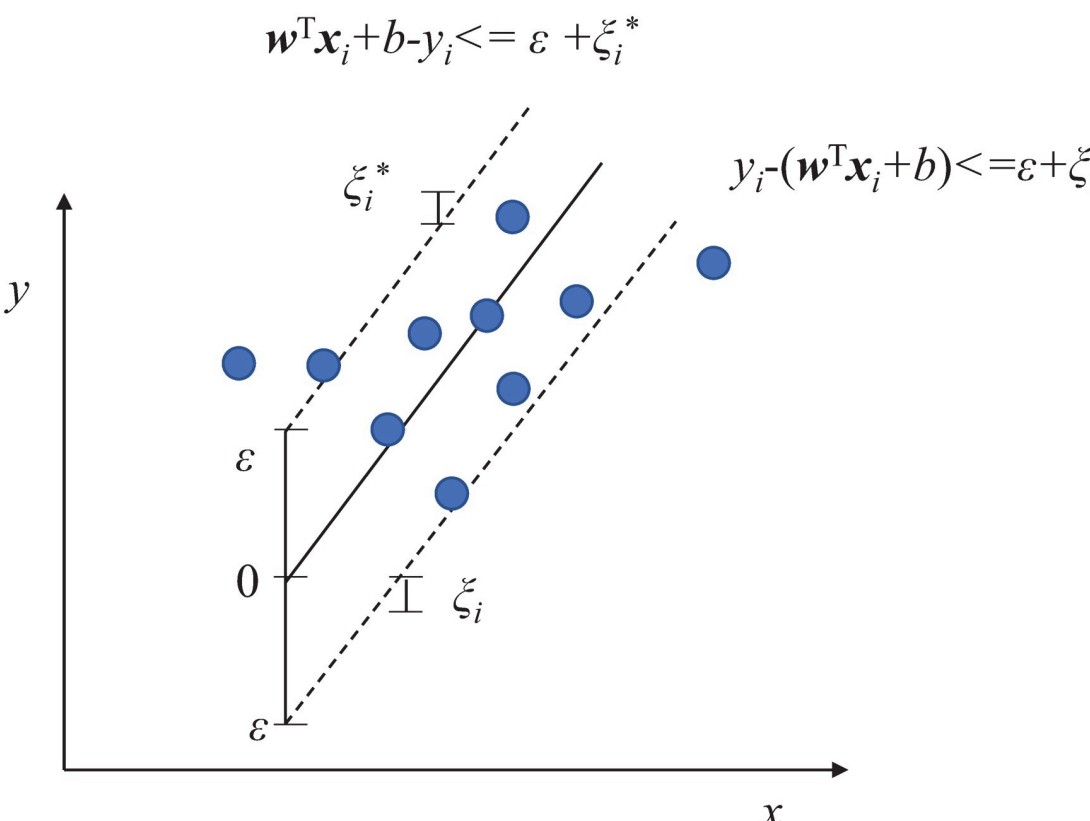

**Fig 5. $\varepsilon$-SVM regression with the $\varepsilon$-insensitive hinge loss, meaning there is no penalty to errors within the $\varepsilon$ margin.**

with the same size of samples from the given training samples based on the re-sampling strategy. Then, for each new training set, each decision tree is trained with the selected features based on recursive partitioning, where a decision tree search is applied for the best split from the selected features. The final output is based on the average of predictions from all the decision trees.

**2.2.4 XGBoost.** XGBoost is also an ensemble model [57]. It employs gradient boosting [58] to group multiple results from the decision tree-based models as the final result. In addition, it uses shrinkage and feature sub-sampling to further reduce the impact of overfitting [59]. XGBoost is suitable in applications that require parallelisation, distributed computing, out-of-core computing, and cache optimisation, which is suitable in real-world applications that have high requirements of computation time and storage memory [60]. The training procedure of XGBoost is depicted in Fig 7. It can be seen from the figure that XGBoost is based on gradient boosting. More specifically, new models (decision trees) are built to predict the errors (residuals) of prior models (from $f_1$ to the current model). Once all the models are obtained, they are integrated together to make the final prediction.

## 3 Experimental results and discussions

In this section, we present the results of the computational experiments conducted based on two body fat datasets—Cases 1 and 2—to validate the effectiveness of feature extraction methods for body fat prediction. Case 1 is based on anthropometric measurements, while Case 2 is based on physical examination and laboratory measurements. We compare four well-known

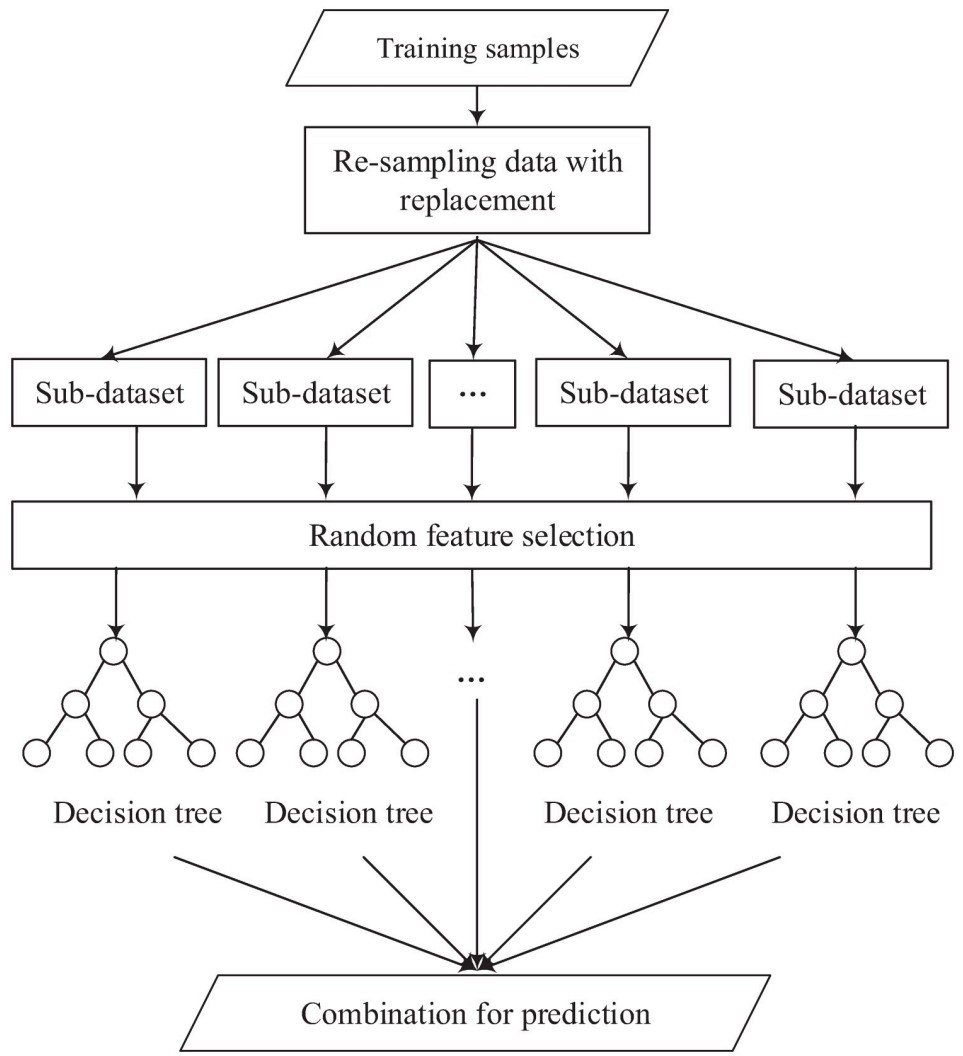

**Fig 6. An example of the RF model.**

machine learning algorithms, the MLP, SVM, RF and XGBoost, with the feature extraction methods used. Specifically, MLP_FA, MLP_PCA and MLP_ICA are the MLP based on FA, PCA and ICA; SVM_FA, SVM_PCA and SVM_ICA are the SVM based on FA, PCA and ICA; RF_FA, RF_PCA and RF_ICA are the RF based FA, PCA and ICA; and XGBoost_FA, XGBoost_PCA and XGBoost_ICA are XGBoost based on FA, PCA and ICA. The programming/development environment was based on Python using scikit-learn, and the experiments were executed on a computer with an i5-6300HQ CPU of 2.30GHz having 16.0 GB RAM.

### 3.1 Performance measures

In this study, we considered five performance measures. Specifically, the *MAE* and *RMSE* were used to evaluate the model's approximation ability, *SD* was used to measure the variability of the errors between the predicted and target values, *MAC* [61] was used to evaluate model robustness, and computation time was used to measure the efficiency. To better evaluate the performance, we randomly shuffled the data and ran the experiments of five-fold cross validation for 20 times, then averaged them to get the final results. The computation time included

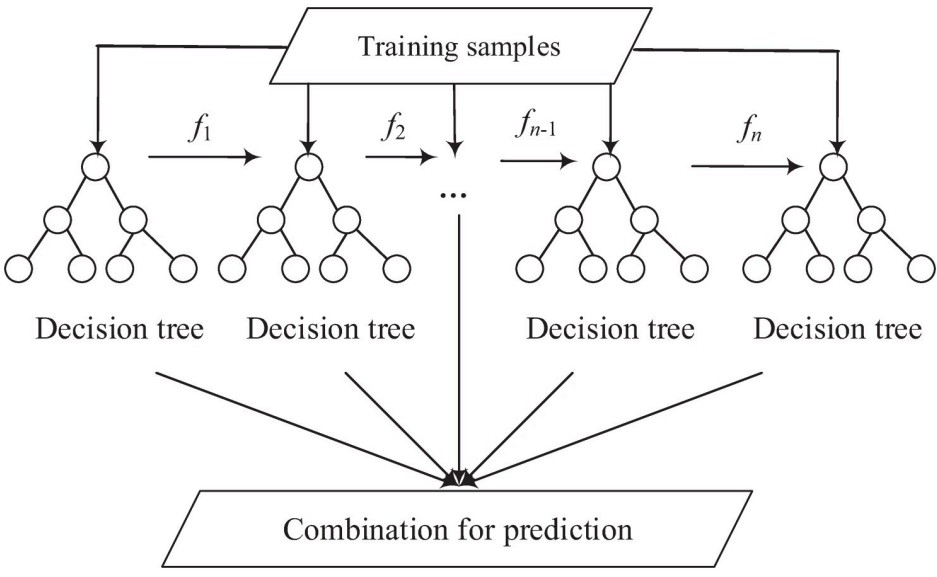

**Fig 7. An example of the XGBoost model.**

the time for feature extraction and 20 runs of five-fold cross validation. Our objective was to minimise the *MAE*, *SD*, *RMSE* and computation time while maximising *MAC*.

$$MAE = \frac{1}{n}\sum_{i=1}^{n}(|y_i^p - y_i^t|),\qquad(4)$$

$$SD = \sqrt{\frac{1}{n}\sum_{i=1}^{n}(e_i - \bar{e})^2},\qquad(5)$$

$$RMSE = \sqrt{\frac{1}{n}\sum_{i=1}^{n}(y_i^p - y_i^t)^2},\qquad(6)$$

$$MAC_{y^p y^t} = \frac{((y^p)^T y^t)^2}{((y^t)^T y^t)((y^p)^T y^p)},\qquad(7)$$

where *n* is the number of samples, $y_i^p$ and $y_i^t$ are prediction and target values of the *i*th sample, respectively, $e_i$ is the *i*th sample's absolute error, $\bar{e}$ is the average of absolute errors, $(y^p)^T y^t$ is the inner product operation for $(y^p)^T$ and $y^t$, and $(y^p)^T$ is the transpose of $y^p$.

### 3.2 Parameter settings

We used the grid search approach with cross validation for parameter selection [62]. The settings used in our experiments, obtained after some tuning process, are listed in Table 1.

A flowchart of different feature extraction methods used for body fat prediction based on *K*-fold cross validation with *N* repeated experiments is given in Fig 8 to further clarify the procedure of our experiments. In the figure, *K* = 5 and *N* = 20; i.e., the experiments were repeated 20 times and each experiment was conducted based on 5-fold cross validation.

**Table 1. Parameter settings for the prediction models, where *#neurons* is the number of neurons, *#iterations* is the maximum number of iterations, *regularisation* is the regularisation parameter, $\sigma^2$ is the variance within the RBF kernel, *#trees* is the number of trees, and *depth* is the maximum depth of the tree.**

|  | Grid search | Optimal parameters |
|---|---|---|
| MLP | *#neurons* = [100, 500, 1000] | *#neurons* = 500 |
|  | *#iterations* = [100, 500, 1000] | *#iterations* = 500 |
| SVM | *regularisation* = [10, 100, 1000] | *regularisation* = 10 |
|  | $1/\sigma^2$ = [0.001, 0.01, 0.1] | $1/\sigma^2$ = 0.001 |
| RF | *#trees* = [10, 100, 1000] | *#trees* = 1000 |
|  | *depth* = [3, 4, 5] | *depth* = 5 |
| XGBoost | *#trees* = [10, 100, 1000] | *#trees* = 100 |
|  | *depth* = [3, 4, 5] | *depth* = 3 |

### 3.3 Case 1: Body fat percentage prediction based on anthropometric measurements

**3.3.1 Data description.** The body fat dataset used in Case 1 contained 252 samples with 13 input features and one output feature. It was downloaded from the StatLib (see http://lib. stat.cmu.edu/datasets/bodyfat). The statistical descriptions of this dataset are provided in

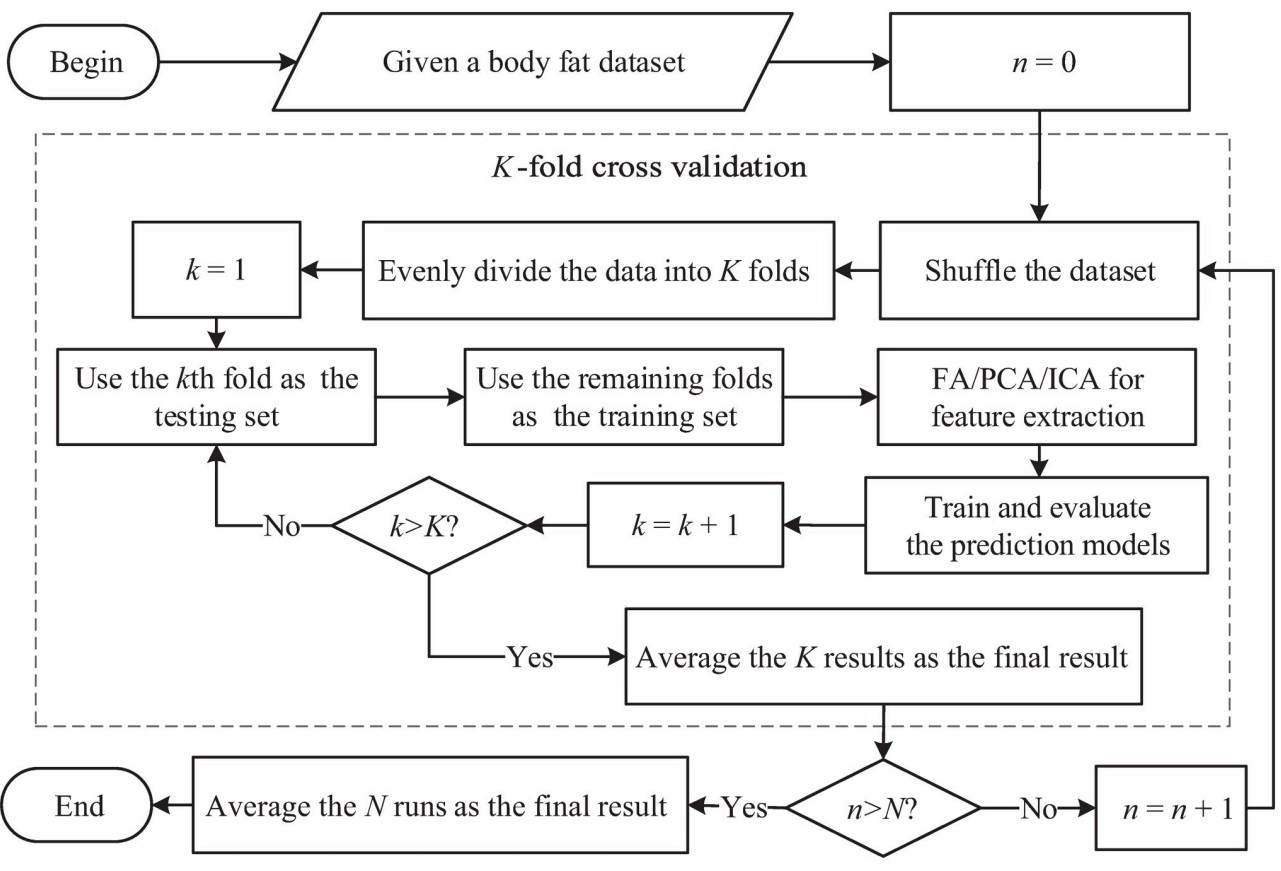

**Fig 8. A flowchart of different feature extraction methods used for body fat prediction based on *K*-fold cross validation with *N* repeated experiments.**

**Table 2. Statistical properties of Case 1's body fat dataset.**

| Variable | Unit | Symbol | Minimum | Maximum | Mean | Standard deviation |
|---|---|---|---|---|---|---|
| Age (years) | years | Age | 22 | 81 | 44.8849 | 12.6020 |
| Weight (lbs) | lbs | Weight | 118.5 | 363.15 | 178.9244 | 29.3892 |
| Height (inches) | inches | Height | 29.5 | 77.75 | 70.1488 | 3.6629 |
| Neck circumference | cm | Neck | 31.1 | 51.2 | 37.9921 | 2.4309 |
| Chest circumference | cm | Chest | 79.3 | 136.2 | 100.8242 | 8.4305 |
| Abdomen 2 circumference | cm | Abdomen | 69.4 | 148.1 | 92.5560 | 10.7831 |
| Hip circumference | cm | Hip | 85 | 147.7 | 99.9048 | 7.1641 |
| Thigh circumference | cm | Thigh | 47.2 | 87.3 | 59.4060 | 5.2500 |
| Knee circumference | cm | Knee | 33 | 49.1 | 38.5905 | 2.4118 |
| Ankle circumference | cm | Ankle | 19.1 | 33.9 | 23.1024 | 1.6949 |
| Biceps (extended) circumference | cm | Biceps | 24.8 | 45 | 32.2734 | 3.0213 |
| Forearm circumference | cm | Forearm | 21 | 34.9 | 28.6639 | 2.0207 |
| Wrist circumference | cm | Wrist | 15.8 | 21.4 | 18.2298 | 0.9336 |
| Body fat percentage | % | Bodyfat% | 0 | 47.5 | 19.1508 | 8.3687 |

Table 2. The input features included age, weight and various body circumference measurements, and the output feature was the body fat percentage.

**3.3.2 Determination of the number of extracted features.** To determine the number of extracted features, we calculated the explained variance for each feature by using scikit-learn [63]. We only selected the principal components that have the largest eigenvalues based on a given threshold (i.e. how much information it contained). The four steps to determine the number of extracted features were as follows: (1) constructing the covariance matrix; (2) decomposing the covariance matrix into its eigenvectors and eigenvalues; (3) sorting the eigenvalues by decreasing order to rank the corresponding eigenvectors; and (4) selecting the $k$ largest eigenvalues such that their cumulative explained variance reached the given threshold. The explained variance ratio for the StatLib dataset is given in Fig 9. Here, the threshold was set to 0.99, which means 99% of the information remained. In this case, six features were extracted from the 13 input features.

**3.3.3 Experiments and results.** Table 3 presents the results obtained by the MLP, SVM, RF and XGBoost for body fat prediction with and without feature extraction. As shown in the table, the SVM, RF and XGBoost perform better than MLP. The performance of SVM and XGBoost is similar, whereas that of RF is the best in terms of accuracy. However, it is clear that, by incorporating feature extraction, the learning models can achieve higher prediction accuracy, stability and robustness in most cases. The XGBoost model with FA feature extraction generated the most precise and stable results, albeit taking longer computation time than the standalone XGBoost. Using the feature extraction method increases the computation time because feature extraction pre-processing also takes time, even though it is more efficient to train the prediction model with less input features. Among all the prediction models, XGBoost with FA for feature extraction shows the best prediction accuracy ($MAE$ = 3.433, $SD$ = 4.188 and $RMSE$ = 4.248), and the SVM with PCA obtained results in the shortest computation time (close to the standalone SVM).

**3.3.4 Statistical analysis based on the Wilcoxon rank-sum test.** Although the results of MLP, SVM and XGBoost presented thus far have shown that the use of feature extraction can improve their performance, statistical analysis is needed to validate whether the differences between the results obtained are statistically significant. In this section, we report the results of statistical tests conducted based on the Wilcoxon rank-sum test [64]. Table 4 shows the

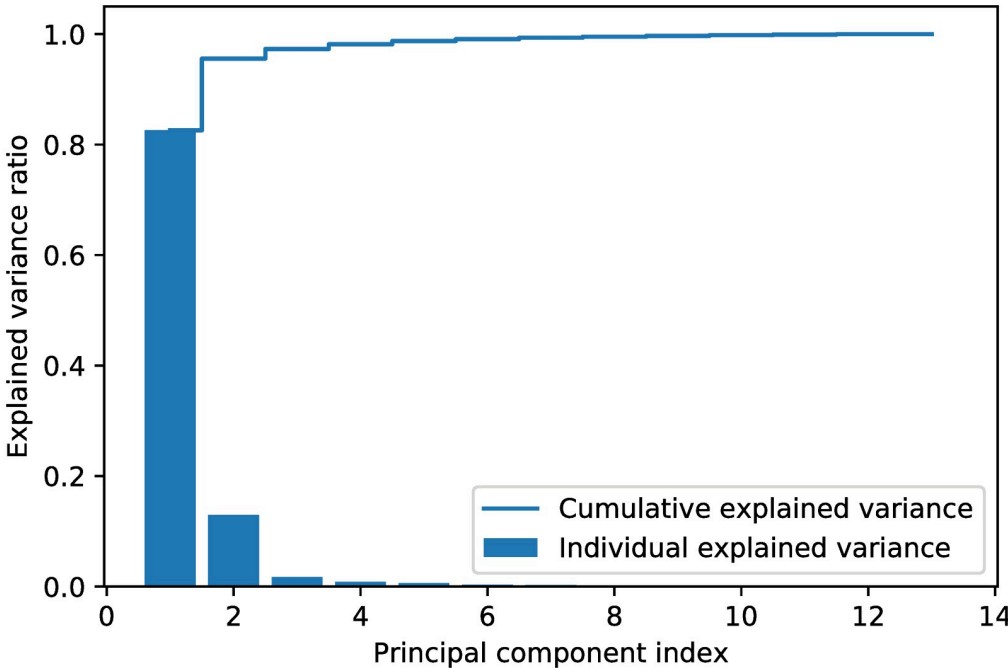

**Fig 9. Explained variance ratio for the StatLib dataset.**

statistical test results based on the 20-run experimental results. As shown in the table, the MLP, SVM and XGBoost and their versions with the feature extraction methods incorporated are significantly different (the $p$-value is less than 0.05). However, the difference between the RF and RF_PCA is not significant. This means the use of feature extraction is effective in improving the performance of MLP, SVM and XGBoost.

**Table 3. Experimental results based on the StatLib dataset (best results are highlighted in bold).**

| Algorithm | MAE | SD | RMSE | MAC | Computation time (s) |
|---|---|---|---|---|---|
| MLP | 6.872 | 8.177 | 8.336 | 0.845 | 150 |
| MLP_FA | 3.764 | 4.547 | 4.606 | 0.952 | 884 |
| MLP_PCA | 3.716 | 4.500 | 4.564 | 0.953 | 378 |
| MLP_ICA | 6.372 | 7.644 | 7.746 | 0.862 | 442 |
| SVM | 3.941 | 4.764 | 4.824 | 0.947 | **9** |
| SVM_FA | 3.740 | 4.529 | 4.603 | 0.952 | 67 |
| SVM_PCA | 3.796 | 4.660 | 4.724 | 0.949 | 10 |
| SVM_ICA | 3.678 | 4.449 | 4.511 | **0.954** | 20 |
| RF | 3.837 | 4.612 | 4.676 | 0.951 | 411 |
| RF_FA | 3.891 | 4.734 | 4.788 | 0.947 | 437 |
| RF_PCA | 3.866 | 4.670 | 4.739 | 0.948 | 374 |
| RF_ICA | 4.007 | 4.814 | 4.891 | 0.945 | 368 |
| XGBoost | 3.945 | 4.758 | 4.829 | 0.947 | 84 |
| XGBoost_FA | **3.433** | **4.188** | **4.248** | 0.949 | 116 |
| XGBoost_PCA | 3.538 | 4.289 | 4.337 | 0.947 | 56 |
| XGBoost_ICA | 3.558 | 4.296 | 4.362 | 0.946 | 74 |

**Table 4. Wilcoxon rank-sum tests for the MLP, SVM, RF, XGBoost, and the use of feature extraction, based on the StatLib dataset in terms of *RMSE* (*p*-values less than 0.05 are highlighted in bold).**

|  | MLP | SVM | RF | XGBoost |
|---|---|---|---|---|
| MLP_FA | **$6.302\times10^{-8}$** |  |  |  |
| MLP_PCA | **$6.302\times10^{-8}$** |  |  |  |
| MLP_ICA | **$6.302\times10^{-8}$** |  |  |  |
| SVM_FA |  | **$3.180\times10^{-7}$** |  |  |
| SVM_PCA |  | **$2.756\times10^{-5}$** |  |  |
| SVM_ICA |  | **$6.302\times10^{-8}$** |  |  |
| RF_FA |  |  | **0.030** |  |
| RF_PCA |  |  | 0.256 |  |
| RF_ICA |  |  | **$6.302\times10^{-8}$** |  |
| XGBoost_FA |  |  |  | **$6.302\times10^{-8}$** |
| XGBoost_PCA |  |  |  | **$6.302\times10^{-8}$** |
| XGBoost_ICA |  |  |  | **$1.329\times10^{-7}$** |

**3.3.5 Prediction performance with more extracted features.** To investigate the impact of having a different number of anthropometric features on the prediction performance, we increased the number of extracted features from 6 (as calculated in Section 3.3.2) to 13 (the total number of input features) in this series of experiments. Tables 5–7 show the results obtained by the MLP, SVM, RF and XGBoost using FA, PCA and ICA, respectively. As shown

**Table 5. Experimental results for the MLP, SVM, RF, and XGBoost, based on the StatLib dataset, with FA feature extraction (best results are highlighted in bold; # means the number of features).**

| # | MLP | | | | SVM | | | | RF | | | | XGBoost | | | |
|---|---|---|---|---|---|---|---|---|---|---|---|---|---|---|---|---|
|  | *MAC* | *SD* | *RMSE* | *MAC* | *MAC* | *SD* | *RMSE* | *MAC* | *MAC* | *SD* | *RMSE* | *MAC* | *MAC* | *SD* | *RMSE* | *MAC* |
| 6 | 3.764 | 4.547 | 4.606 | 0.952 | 3.740 | 4.529 | 4.603 | **0.952** | 3.891 | 4.734 | 4.788 | 0.947 | **3.433** | **4.188** | **4.248** | 0.949 |
| 7 | 3.719 | 4.474 | 4.539 | **0.953** | 3.728 | 4.483 | 4.542 | 0.953 | 3.939 | 4.783 | 4.833 | 0.946 | **3.511** | **4.252** | **4.296** | 0.948 |
| 8 | 3.718 | 4.449 | 4.504 | 0.954 | 3.665 | 4.424 | 4.501 | **0.954** | 3.905 | 4.725 | 4.777 | 0.947 | **3.463** | **4.179** | **4.231** | 0.949 |
| 9 | 3.674 | 4.396 | 4.447 | **0.955** | 3.627 | 4.403 | 4.466 | 0.955 | 3.946 | 4.769 | 4.832 | 0.946 | **3.463** | **4.163** | **4.218** | 0.950 |
| 10 | 3.672 | 4.381 | 4.433 | 0.955 | 3.542 | 4.344 | 4.408 | **0.956** | 3.915 | 4.748 | 4.791 | 0.947 | **3.460** | **4.160** | **4.217** | 0.950 |
| 11 | 3.653 | 4.356 | 4.436 | **0.956** | 3.556 | 4.399 | 4.458 | 0.955 | 3.926 | 4.772 | 4.827 | 0.946 | **3.445** | **4.143** | **4.210** | 0.951 |
| 12 | 3.634 | 4.347 | 4.414 | **0.956** | 3.517 | 4.356 | 4.428 | 0.956 | 3.979 | 4.803 | 4.890 | 0.945 | **3.464** | **4.196** | **4.247** | 0.949 |
| 13 | 3.671 | 4.404 | 4.471 | **0.955** | 3.647 | 4.419 | 4.484 | 0.955 | 3.934 | 4.771 | 4.819 | 0.946 | **3.462** | **4.152** | **4.202** | 0.950 |

**Table 6. Experimental results for the MLP, SVM, RF, and XGBoost, based on the StatLib dataset, with PCA feature extraction (best results are highlighted in bold; # means the number of features).**

| # | MLP | | | | SVM | | | | RF | | | | XGBoost | | | |
|---|---|---|---|---|---|---|---|---|---|---|---|---|---|---|---|---|
|  | *MAC* | *SD* | *RMSE* | *MAC* | *MAC* | *SD* | *RMSE* | *MAC* | *MAC* | *SD* | *RMSE* | *MAC* | *MAC* | *SD* | *RMSE* | *MAC* |
| 6 | 3.716 | 4.500 | 4.564 | **0.953** | 3.796 | 4.660 | 4.724 | 0.949 | 3.866 | 4.670 | 4.739 | 0.948 | **3.538** | **4.289** | **4.337** | 0.947 |
| 7 | 3.727 | 4.504 | 4.557 | **0.953** | 3.820 | 4.671 | 4.728 | 0.949 | 3.901 | 4.722 | 4.782 | 0.947 | **3.511** | **4.237** | **4.287** | 0.948 |
| 8 | 3.729 | 4.503 | 4.556 | **0.953** | 3.850 | 4.693 | 4.764 | 0.948 | 3.914 | 4.774 | 4.823 | 0.946 | **3.558** | **4.277** | **4.336** | 0.947 |
| 9 | 3.754 | 4.515 | 4.580 | **0.952** | 3.822 | 4.663 | 4.722 | 0.949 | 3.916 | 4.744 | 4.805 | 0.947 | **3.530** | **4.286** | **4.324** | 0.947 |
| 10 | 3.737 | 4.501 | 4.543 | **0.953** | 3.807 | 4.662 | 4.729 | 0.949 | 3.946 | 4.778 | 4.835 | 0.946 | **3.520** | **4.279** | **4.320** | 0.947 |
| 11 | 3.703 | 4.445 | 4.518 | **0.954** | 3.789 | 4.623 | 4.687 | 0.950 | 3.919 | 4.748 | 4.817 | 0.946 | **3.493** | **4.222** | **4.273** | 0.948 |
| 12 | 3.716 | 4.475 | 4.546 | **0.953** | 3.781 | 4.629 | 4.693 | 0.950 | 3.958 | 4.825 | 4.864 | 0.944 | **3.485** | **4.229** | **4.290** | 0.948 |
| 13 | 3.635 | 4.401 | 4.456 | **0.955** | 3.777 | 4.618 | 4.683 | 0.950 | 3.981 | 4.818 | 4.893 | 0.945 | **3.438** | **4.149** | **4.205** | 0.950 |

**Table 7. Experimental results for the MLP, SVM, RF, and XGBoost, based on the StatLib dataset, with ICA feature extraction (best results are highlighted in bold; # means the number of features).**

| # | MLP | | | | SVM | | | | RF | | | | XGBoost | | | |
|---|------|------|------|------|------|------|------|------|------|------|------|------|------|------|------|------|
| | MAC | SD | RMSE | MAC | MAC | SD | RMSE | MAC | MAC | SD | RMSE | MAC | MAC | SD | RMSE | MAC |
| 6 | 6.372 | 7.644 | 7.746 | 0.862 | 3.678 | 4.449 | 4.511 | **0.954** | 4.007 | 4.814 | 4.891 | 0.945 | **3.558** | **4.296** | **4.362** | 0.946 |
| 7 | 6.349 | 7.533 | 7.687 | 0.868 | 3.730 | 4.536 | 4.592 | **0.952** | 4.168 | 5.032 | 5.105 | 0.940 | **3.588** | **4.366** | **4.430** | 0.944 |
| 8 | 6.305 | 7.507 | 7.630 | 0.868 | 3.791 | 4.648 | 4.708 | **0.949** | 4.335 | 5.220 | 5.286 | 0.935 | **3.676** | **4.492** | **4.546** | 0.941 |
| 9 | 6.271 | 7.515 | 7.596 | 0.865 | 3.792 | 4.621 | 4.689 | **0.950** | 4.399 | 5.293 | 5.345 | 0.933 | **3.693** | **4.503** | **4.558** | 0.941 |
| 10 | 6.286 | 7.493 | 7.624 | 0.866 | 3.799 | 4.606 | 4.670 | **0.950** | 4.410 | 5.325 | 5.393 | 0.932 | **3.701** | **4.492** | **4.574** | 0.941 |
| 11 | 6.275 | 7.500 | 7.611 | 0.866 | 3.730 | 4.508 | 4.573 | **0.953** | 4.418 | 5.325 | 5.389 | 0.933 | **3.626** | **4.408** | **4.463** | 0.944 |
| 12 | 6.205 | 7.448 | 7.541 | 0.867 | 3.754 | 4.556 | 4.618 | **0.952** | 4.586 | 5.491 | 5.575 | 0.929 | **3.657** | **4.499** | **4.556** | 0.942 |
| 13 | 6.234 | 7.407 | 7.543 | 0.871 | 3.714 | 4.496 | 4.563 | **0.953** | 4.563 | 5.496 | 5.548 | 0.928 | **3.612** | **4.438** | **4.494** | 0.942 |

in Tables 5–7, in most cases, the accuracy (*RMSE* and *MAE*) and stability (*SD* and *MAC*) were not necessarily enhanced by extracting more features as the inputs of the learning models. Among the models being compared, XGBoost-FA performs the best for predicting the body fat percentage in terms of *MAE*, *RMSE*, *SD* and *MAC*, which means it is able to predict the body fat percentage with the highest accuracy and stability on the StatLib dataset.

It is critical to reduce the number of dimensions when the data size or the number of dimensions is large (big data scenarios). In addition, the prediction models with PCA outperform the corresponding versions with ICA in terms of all the metrics used. This might be due to the Gaussian distribution of the body fat dataset since PCA can process the Gaussian distribution data while ICA cannot.

Fig 10 depicts the comparative experimental results of the computation time for the MLP, SVM, RF and XGBoost using FA, PCA and ICA, respectively. The results show that XGBoost with FA is the fastest among the compared methods. Fig 10 also reveals that in some cases, the computation time increases with more features, which further highlights the importance of

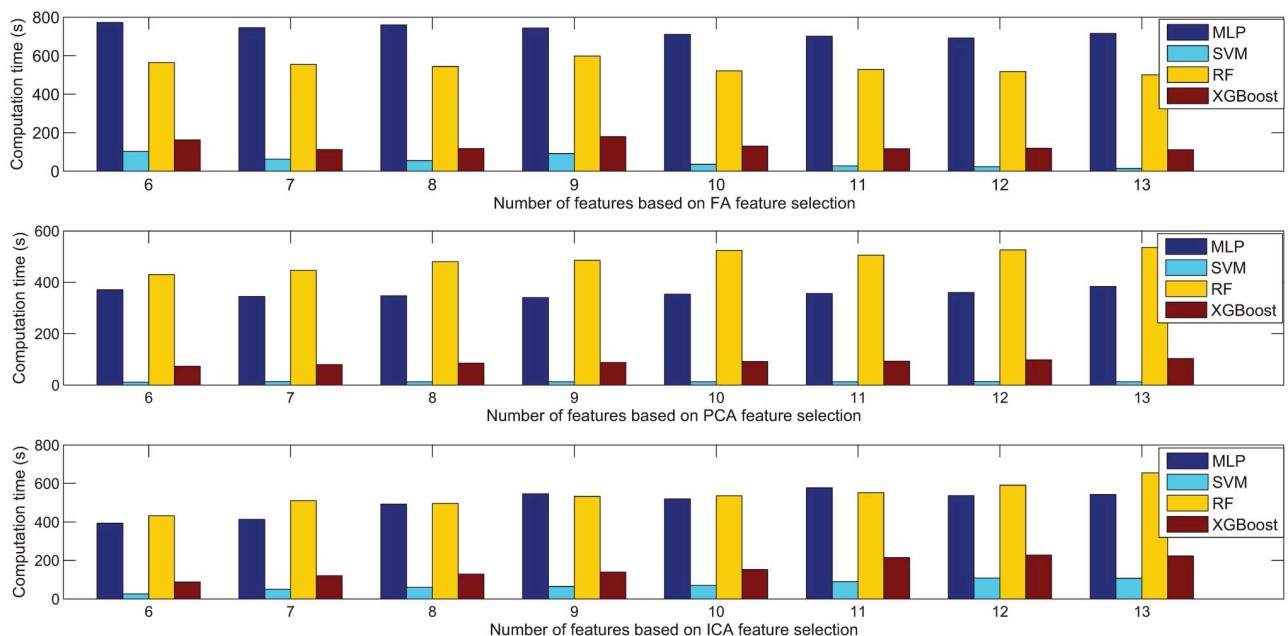

**Fig 10. Comparison results in terms of computation time based on FA, PCA and ICA feature extraction for the StatLib dataset.**

feature extraction in improving the efficiency. The computation time includes the time for feature extraction and 20 runs of five-fold cross validation, which means that when a different number of features are extracted, the time for feature extraction may also differ.

### 3.4 Case 2: Body fat percentage prediction based on physical examination and laboratory measurements

**3.4.1 Data description.** The body fat dataset used in Case 2 was downloaded from the NHANES (see https://www.cdc.gov/nchs/nhanes/index.htm). The data were pre-processed as in [65] by (1) combining *DEMO*, *LAB11*, *LAB18*, *LAB25*, *BMX*, and *BIX* files into one dataset, (2) keeping data on male adults (age > 18); and (3) removing samples with missing values. After pre-processing, 862 samples with 39 features were obtained. These features and their statistical descriptions are provided in Table 8.

**3.4.2 Determination of the number of extracted features.** We ran the same experiment as in Section 3.3.2 to determine the number of extracted features. The explained variance ratio for the NHANES dataset is given in Fig 11. With the threshold set to 0.99, 12 features were extracted from the 38 input features.

**3.4.3 Experiment results.** Table 9 presents results obtained through the MLP, SVM, RF and XGBoost for body fat prediction with and without feature extraction. These results are consistent with those shown in Table 3, and show that ensemble models such as XGBoost performs better than the MLP and SVM. Similarly, results show that incorporating feature extraction into the prediction models enhances the body fat prediction accuracy. The XGBoost model with PCA feature extraction generated the most precise and stable results, as well as shorter computation time than the standalone XGBoost.

**3.4.4 Statistical analysis based on the Wilcoxon rank-sum test.** Table 10 presents statistical test results between the experimental results with and without feature extraction pre-processing. As shown in the table, the MLP, SVM, RF and XGBoost and their versions that use feature extraction are significantly different (the *p*-value is less than 0.05). This means the use of feature extraction methods are effective in improving the performance of MLP, SVM and XGBoost, but not that of RF (the performance of RF_FA, RF_PCA and RF_ICA is less than that of RF in Table 9).

**3.4.5 Prediction performance with more extracted features.** To evaluate the prediction performance on increasing the number of extracted features, we conducted experiments in which the number of features used ranged from 12 (as calculated in Section 3.4.2) to 38 (the total number of input features).

Tables 11–13 show the results obtained from the MLP, SVM, RF and XGBoost by using FA, PCA and ICA for feature extraction, respectively. From the tables, we can observe that with more features extracted, the prediction models can be further improved using feature extraction methods. Table 11 shows that XGBoost based on FA feature extraction has the best prediction accuracy (3.713, 4.707 and 4.728 in terms of (*MAE*, *SD*, *RMSE*) using 38 features. However, it performs satisfactorily using 24 features (3.772, 4.783, 4.803), which is more feasible in real applications. As shown in Table 12, the MLP has the best performance using 35 features. It has improved (from 4.160, 5.230, 5.250 and 0.948 to 3.621, 4.618, 4.647 and 0.960) in terms of *MAE*, *SD*, *RMSE* and *MAC*. As Table 13 shows, XGBoost outperforms the other models in comparison with the use of different number of features. Its best result is 3.805, 4.818, 4.840 and 0.955 in terms of *MAE*, *SD*, *RMSE* and *MAC* based on 24 extracted features. The results with 38 features are used as the baseline. Analysing the results from Tables 11–13 reveals that the MLP, SVM, RF, and XGBoost with feature extraction performed similarly or better than their corresponding baselines in terms of all metrics with only half the features (19

**Table 8. Statistical properties of Case 2's body fat dataset.** More details can be found at https://www.cdc.gov/nchs/nhanes/index.htm.

| Variable | Unit | Symbol | Minimum | Maximum | Mean | Standard deviation |
|---|---|---|---|---|---|---|
| Segmented neutrophils number | | ANC | 1.2 | 9.9 | 4.011 | 1.5443 |
| Basophils number | | ABC | 0 | 0.2 | 0.0311 | 0.0468 |
| Lymphocyte number | | ALC | 0.3 | 5.3 | 2.0561 | 0.6086 |
| Monocyte number | | AMC | 0.1 | 1.6 | 0.5812 | 0.1828 |
| Eosinophils number | | AEC | 0 | 2.1 | 0.206 | 0.1736 |
| Red cell count SI | | RBC | 3.43 | 6.78 | 5.1373 | 0.389 |
| Hemoglobin | (g/dL)*10 | HGB | 113 | 183 | 155.3677 | 9.953 |
| Hematocrit | % / 100 | HCT | 0.355 | 0.547 | 0.461 | 0.028 |
| Mean cell volume | fL | MCV | 65.1 | 108.6 | 89.9342 | 4.4912 |
| Mean cell hemoglobin | pg | MCH | 20.9 | 37.4 | 30.3227 | 1.7578 |
| Mean cell volume | fL * 10 | MCHC | 310 | 360 | 337.0534 | 7.49 |
| Red cell distribution width | % | RDW | 11 | 18.8 | 12.4017 | 0.7036 |
| Platelet count | (%) SI | PLT | 11 | 491 | 251.3631 | 55.2816 |
| Mean platelet volume | fL | MPV | 6.1 | 11.8 | 8.3609 | 0.8788 |
| Sodium | mmol/L | SNA | 129.9 | 146.4 | 139.7056 | 2.3272 |
| Potassium | mmol/L | SK | 3.11 | 5.36 | 4.1586 | 0.3065 |
| Chloride | mmol/L | SCL | 92.4 | 112.3 | 102.0905 | 2.8116 |
| Calcium, total | mmol/L | SCA | 2.125 | 2.7 | 2.3791 | 0.0912 |
| Phosphorus | mmol/L | SP | 0.549 | 2.357 | 1.1111 | 0.164 |
| Bilirubin, total | umol/L | STB | 3.4 | 63.3 | 11.6911 | 5.9454 |
| Bicarbonate | mmol/L | BIC | 17 | 32 | 24.1717 | 2.2766 |
| Glucose | mmol/L | GLU | 3.22 | 31.141 | 5.1061 | 1.5484 |
| Iron | umol/L | IRN | 3.94 | 46.39 | 17.9708 | 6.5508 |
| LDH | U/L | LDH | 45 | 578 | 151.9362 | 34.5774 |
| Protein, total | g/L | STP | 64 | 96 | 77.0151 | 4.3629 |
| Uric acid | umol/L | SUA | 172.5 | 642.4 | 354.5239 | 71.593 |
| Albumin | g/L | SAL | 34 | 57 | 46.8329 | 2.8720 |
| Triglycerides | mmol/L | TRI | 0.282 | 11.595 | 1.5610 | 1.2548 |
| Blood urea nitrogen | mmol/L | BUN | 1.4 | 15 | 4.9233 | 1.2775 |
| Creatinine | umol/L | SCR | 35.4 | 901.7 | 72.4034 | 31.6012 |
| Cholesterol, total | mmol/L | STC | 1.68 | 9.72 | 4.9024 | 1.0921 |
| AST | U/L | AST | 9 | 827 | 29.0325 | 37.1348 |
| ALT | U/L | ALT | 7 | 1163 | 34.7738 | 47.1822 |
| GGT | U/L | GGT | 7 | 698 | 37.4849 | 47.624 |
| Alkaline phosphotase | U/L | ALP | 30 | 271 | 84.2541 | 25.5154 |
| Weight | kg | WT | 42.7 | 138.1 | 81.9086 | 17.2013 |
| Standing height | cm | HT | 152.3 | 201.3 | 174.3531 | 7.8856 |
| Waist circumference | cm | WC | 62.4 | 147.7 | 93.3414 | 13.6034 |
| Estimated percent body fat | % | BFP | 4 | 61.8 | 24.1874 | 7.5771 |

features). This shows the potential of greatly improving the efficiency in real-world applications. In addition, analysis reveals that PCA is more suitable for extracting features for the body fat dataset than ICA. The reason could be that this body fat dataset has a Gaussian distribution and PCA is better suited for Gaussian-distribution data whereas ICA is better suited for non-Gaussian distribution data.

Among the three feature extraction algorithms, PCA is the most effective one for this dataset. It greatly improves the performance of the prediction models being compared. In addition,

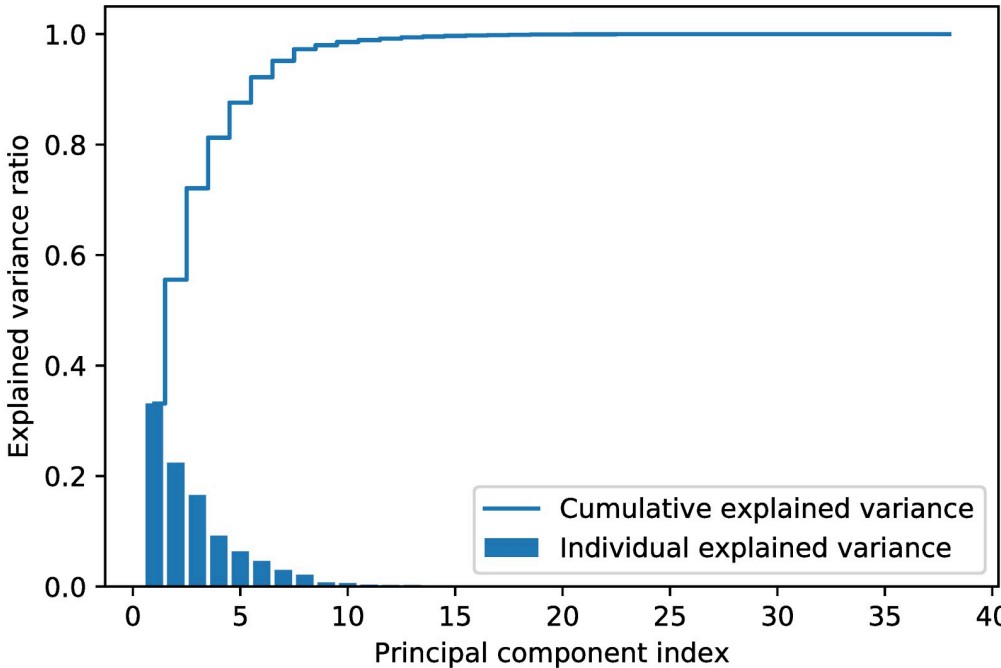

**Fig 11. Explained variance ratio for the NHANES dataset.**

Fig 12 depicts the comparative experimental results of computation time for the MLP, SVM, RF and XGBoost with different number of features extracted from FA, PCA and ICA. As shown in the figure, for each prediction model, there is a trend that with more features used, more time is needed. The prediction models ordered by computation time from the most time-consuming to the most efficient are the MLP, RF, XGBoost and SVM.

**Table 9. Experimental results based on the NHANES dataset (best results are highlighted in bold).**

| Algorithm | MAE | SD | RMSE | MAC | Computation time (s) |
|---|---|---|---|---|---|
| MLP | 5.088 | 6.394 | 6.434 | 0.936 | 689 |
| MLP_FA | 4.727 | 6.015 | 6.038 | 0.943 | 3360 |
| MLP_PCA | 4.160 | 5.230 | 5.250 | 0.948 | 2684 |
| MLP_ICA | 4.919 | 6.265 | 6.288 | 0.939 | 1206 |
| SVM | 6.022 | 7.542 | 7.572 | 0.911 | 70 |
| SVM_FA | 6.210 | 8.030 | 8.060 | 0.887 | 563 |
| SVM_PCA | 4.837 | 6.058 | 6.081 | 0.929 | 63 |
| SVM_ICA | 4.705 | 6.203 | 6.225 | 0.939 | **28** |
| RF | 4.554 | 5.730 | 5.746 | 0.949 | 1479 |
| RF_FA | 4.822 | 6.044 | 6.064 | 0.943 | 1068 |
| RF_PCA | 4.696 | 5.856 | 5.877 | 0.946 | 542 |
| RF_ICA | 4.706 | 5.889 | 5.905 | 0.946 | 549 |
| XGBoost | 4.592 | 5.780 | 5.802 | 0.948 | 584 |
| XGBoost_FA | 4.169 | 5.255 | 5.276 | 0.946 | 656 |
| XGBoost_PCA | **4.021** | **5.07** | **5.089** | **0.950** | 178 |
| XGBoost_ICA | 4.039 | 5.081 | 5.096 | **0.950** | 183 |

**Table 10. Wilcoxon rank-sum tests for the MLP, SVM, RF, XGBoost, and the use of feature extraction, based on the NHANES dataset in terms of *RMSE* (*p*-values less than 0.05 are highlighted in bold).**

| | MLP | SVM | RF | XGBoost |
|---|---|---|---|---|
| MLP_FA | **6.302×10⁻⁸** | | | |
| MLP_PCA | **6.302×10⁻⁸** | | | |
| MLP_ICA | **4.229×10⁻⁷** | | | |
| SVM_FA | | **6.302×10⁻⁸** | | |
| SVM_PCA | | **6.302×10⁻⁸** | | |
| SVM_ICA | | **6.302×10⁻⁸** | | |
| RF_FA | | | **6.302×10⁻⁸** | |
| RF_PCA | | | **1.473×10⁻⁶** | |
| RF_ICA | | | **5.509×10⁻⁶** | |
| XGBoost_FA | | | | **6.302×10⁻⁸** |
| XGBoost_PCA | | | | **6.302×10⁻⁸** |
| XGBoost_ICA | | | | **6.302×10⁻⁸** |

**Table 11. Experimental results for the MLP, SVM, RF, and XGBoost, based on the NHANES dataset, with FA feature extraction (best results are highlighted in bold; # means the number of features).**

| # | MLP | | | | SVM | | | | RF | | | | XGBoost | | | |
|---|---|---|---|---|---|---|---|---|---|---|---|---|---|---|---|---|
| | *MAC* | *SD* | *RMSE* | *MAC* | *MAC* | *SD* | *RMSE* | *MAC* | *MAC* | *SD* | *RMSE* | *MAC* | *MAC* | *SD* | *RMSE* | *MAC* |
| 12 | 4.727 | 6.015 | 6.038 | 0.943 | 6.210 | 8.030 | 8.060 | 0.887 | 4.822 | 6.044 | 6.064 | 0.943 | **4.169** | **5.255** | **5.276** | **0.946** |
| 13 | 4.590 | 5.810 | 5.827 | 0.947 | 5.525 | 7.057 | 7.086 | 0.909 | 4.711 | 5.884 | 5.907 | 0.946 | **4.021** | **5.062** | **5.074** | **0.950** |
| 14 | 4.599 | 5.810 | 5.832 | 0.947 | 5.248 | 6.619 | 6.638 | 0.917 | 4.717 | 5.884 | 5.901 | 0.946 | **4.034** | **5.073** | **5.092** | **0.950** |
| 15 | 4.585 | 5.796 | 5.814 | 0.948 | 5.043 | 6.336 | 6.358 | 0.923 | 4.739 | 5.955 | 5.981 | 0.944 | **4.031** | **5.078** | **5.103** | **0.950** |
| 16 | 4.545 | 5.730 | 5.748 | 0.949 | 4.697 | 5.862 | 5.882 | 0.934 | 4.711 | 5.905 | 5.924 | 0.945 | **4.021** | **5.070** | **5.088** | **0.950** |
| 17 | 4.539 | 5.745 | 5.762 | 0.948 | 4.542 | 5.662 | 5.679 | 0.938 | 4.748 | 5.937 | 5.964 | 0.945 | **4.017** | **5.045** | **5.069** | **0.950** |
| 18 | 4.304 | 5.436 | 5.454 | 0.954 | 4.198 | 5.270 | 5.292 | 0.947 | 4.573 | 5.721 | 5.742 | 0.948 | **3.855** | **4.860** | **4.879** | **0.954** |
| 19 | 4.262 | 5.396 | 5.410 | 0.954 | 4.130 | 5.194 | 5.210 | 0.948 | 4.556 | 5.701 | 5.723 | 0.949 | **3.837** | **4.844** | **4.859** | **0.955** |
| 20 | 4.250 | 5.401 | 5.419 | 0.954 | 4.076 | 5.107 | 5.127 | 0.950 | 4.558 | 5.722 | 5.747 | 0.948 | **3.802** | **4.802** | **4.820** | **0.955** |
| 21 | 4.189 | 5.345 | 5.360 | **0.955** | 4.039 | 5.106 | 5.128 | 0.950 | 4.592 | 5.736 | 5.749 | 0.948 | **3.809** | **4.824** | **4.840** | 0.955 |
| 22 | 4.150 | 5.279 | 5.295 | **0.956** | 3.955 | 5.011 | 5.027 | 0.951 | 4.597 | 5.739 | 5.763 | 0.948 | **3.784** | **4.788** | **4.809** | 0.955 |
| 23 | 4.161 | 5.299 | 5.319 | **0.956** | 3.924 | 4.982 | 5.009 | 0.952 | 4.600 | 5.737 | 5.756 | 0.948 | **3.792** | **4.767** | **4.785** | 0.956 |
| 24 | 4.155 | 5.297 | 5.315 | **0.956** | 3.923 | 4.999 | 5.014 | 0.952 | 4.606 | 5.760 | 5.781 | 0.948 | **3.772** | **4.783** | **4.803** | 0.956 |
| 25 | 4.148 | 5.294 | 5.311 | **0.956** | 3.897 | 4.954 | 4.980 | 0.952 | 4.601 | 5.762 | 5.781 | 0.948 | **3.809** | **4.790** | **4.807** | 0.955 |
| 26 | 4.145 | 5.291 | 5.307 | **0.956** | 3.897 | 4.973 | 4.991 | 0.952 | 4.608 | 5.753 | 5.772 | 0.948 | **3.800** | **4.782** | **4.802** | 0.955 |
| 27 | 4.136 | 5.274 | 5.291 | **0.956** | 3.890 | 4.951 | 4.975 | 0.952 | 4.625 | 5.768 | 5.796 | 0.948 | **3.798** | **4.787** | **4.810** | 0.955 |
| 28 | 4.141 | 5.291 | 5.302 | **0.956** | 3.897 | 4.963 | 4.980 | 0.952 | 4.629 | 5.775 | 5.790 | 0.948 | **3.787** | **4.782** | **4.798** | 0.955 |
| 29 | 4.136 | 5.263 | 5.287 | **0.956** | 3.898 | 4.956 | 4.973 | 0.952 | 4.615 | 5.753 | 5.772 | 0.948 | **3.832** | **4.822** | **4.841** | 0.955 |
| 30 | 4.118 | 5.259 | 5.276 | **0.957** | 3.913 | 4.994 | 5.015 | 0.952 | 4.614 | 5.764 | 5.781 | 0.948 | **3.838** | **4.824** | **4.843** | 0.955 |
| 31 | 4.101 | 5.233 | 5.254 | **0.957** | 3.890 | 4.963 | 4.982 | 0.952 | 4.605 | 5.761 | 5.779 | 0.948 | **3.814** | **4.817** | **4.834** | 0.955 |
| 32 | 4.098 | 5.225 | 5.240 | **0.957** | 3.902 | 4.978 | 5.002 | 0.952 | 4.607 | 5.755 | 5.774 | 0.948 | **3.799** | **4.791** | **4.809** | 0.955 |
| 33 | 4.112 | 5.243 | 5.260 | **0.957** | 3.884 | 4.968 | 4.982 | 0.952 | 4.633 | 5.793 | 5.821 | 0.947 | **3.797** | **4.803** | **4.822** | 0.955 |
| 34 | 4.092 | 5.226 | 5.246 | **0.957** | 3.901 | 4.985 | 5.000 | 0.952 | 4.626 | 5.780 | 5.804 | 0.947 | **3.793** | **4.809** | **4.825** | 0.955 |
| 35 | 4.109 | 5.233 | 5.252 | **0.957** | 3.898 | 4.984 | 5.002 | 0.952 | 4.631 | 5.792 | 5.811 | 0.947 | **3.801** | **4.801** | **4.821** | 0.955 |
| 36 | 4.105 | 5.231 | 5.251 | **0.957** | 3.887 | 4.966 | 4.981 | 0.952 | 4.629 | 5.776 | 5.798 | 0.948 | **3.813** | **4.809** | **4.831** | 0.955 |
| 37 | 4.110 | 5.236 | 5.255 | **0.957** | 3.880 | 4.953 | 4.970 | 0.953 | 4.632 | 5.782 | 5.801 | 0.947 | **3.777** | **4.774** | **4.791** | 0.955 |
| 38 | 4.120 | 5.233 | 5.256 | **0.957** | 3.909 | 4.965 | 4.986 | 0.952 | 4.526 | 5.649 | 5.665 | 0.950 | **3.713** | **4.707** | **4.728** | 0.957 |

**Table 12. Experimental results for the MLP, SVM, RF, and XGBoost, based on the NHANES dataset, with PCA feature extraction (best results are highlighted in bold; # means the number of features).**

| # | MLP | | | | SVM | | | | RF | | | | XGBoost | | | |
|---|------|------|------|------|------|------|------|------|------|------|------|------|------|------|------|------|
| | MAC | SD | RMSE | MAC | MAC | SD | RMSE | MAC | MAC | SD | RMSE | MAC | MAC | SD | RMSE | MAC |
| 12 | 4.160 | 5.230 | 5.250 | 0.948 | 4.837 | 6.058 | 6.081 | 0.929 | 4.696 | 5.856 | 5.877 | 0.946 | **4.021** | **5.07** | **5.089** | **0.950** |
| 13 | 4.056 | 5.119 | 5.139 | 0.950 | 4.831 | 6.046 | 6.065 | 0.929 | 4.668 | 5.826 | 5.848 | 0.947 | **4.010** | **5.06** | **5.077** | **0.950** |
| 14 | 4.057 | 5.108 | 5.126 | 0.950 | 4.841 | 6.070 | 6.093 | 0.929 | 4.661 | 5.815 | 5.837 | 0.947 | **4.026** | **5.06** | **5.086** | **0.950** |
| 15 | **4.000** | **5.052** | **5.073** | **0.951** | 4.852 | 6.077 | 6.099 | 0.929 | 4.694 | 5.863 | 5.883 | 0.946 | 4.028 | 5.069 | 5.088 | 0.950 |
| 16 | **3.982** | **5.018** | **5.033** | **0.952** | 4.831 | 6.049 | 6.071 | 0.929 | 4.677 | 5.810 | 5.833 | 0.947 | 4.023 | 5.062 | 5.078 | 0.950 |
| 17 | **3.816** | **4.807** | 4.829 | **0.955** | 4.827 | 6.057 | 6.074 | 0.929 | 4.545 | 5.678 | 5.694 | 0.949 | 3.835 | 4.809 | **4.828** | 0.955 |
| 18 | **3.743** | **4.736** | **4.755** | **0.957** | 4.830 | 6.050 | 6.068 | 0.929 | 4.514 | 5.634 | 5.656 | 0.950 | 3.850 | 4.819 | 4.842 | 0.955 |
| 19 | **3.716** | **4.688** | **4.707** | **0.958** | 4.828 | 6.035 | 6.061 | 0.929 | 4.524 | 5.657 | 5.673 | 0.950 | 3.804 | 4.797 | 4.816 | 0.955 |
| 20 | **3.697** | **4.693** | **4.714** | **0.958** | 4.849 | 6.072 | 6.093 | 0.928 | 4.520 | 5.653 | 5.670 | 0.950 | 3.786 | 4.773 | 4.795 | 0.956 |
| 21 | **3.677** | **4.669** | **4.689** | **0.958** | 4.831 | 6.036 | 6.059 | 0.929 | 4.530 | 5.649 | 5.666 | 0.950 | 3.769 | 4.759 | 4.775 | 0.956 |
| 22 | **3.663** | **4.655** | **4.675** | **0.958** | 4.833 | 6.054 | 6.079 | 0.929 | 4.524 | 5.645 | 5.663 | 0.950 | 3.765 | 4.768 | 4.783 | 0.956 |
| 23 | **3.645** | **4.662** | **4.688** | **0.960** | 4.845 | 6.077 | 6.099 | 0.928 | 4.539 | 5.673 | 5.689 | 0.949 | 3.747 | 4.731 | 4.751 | 0.957 |
| 24 | **3.665** | **4.642** | **4.672** | **0.959** | 4.828 | 6.048 | 6.066 | 0.929 | 4.524 | 5.659 | 5.671 | 0.950 | 3.722 | 4.704 | 4.721 | 0.957 |
| 25 | **3.661** | **4.658** | **4.687** | **0.960** | 4.856 | 6.091 | 6.116 | 0.928 | 4.515 | 5.671 | 5.688 | 0.949 | 3.719 | 4.710 | 4.728 | 0.957 |
| 26 | **3.652** | **4.653** | **4.679** | **0.960** | 4.812 | 6.023 | 6.049 | 0.930 | 4.532 | 5.652 | 5.675 | 0.950 | 3.744 | 4.730 | 4.746 | 0.956 |
| 27 | **3.648** | **4.667** | **4.688** | **0.959** | 4.838 | 6.075 | 6.098 | 0.928 | 4.502 | 5.626 | 5.646 | 0.950 | 3.751 | 4.729 | 4.749 | 0.956 |
| 28 | **3.650** | **4.644** | **4.667** | **0.960** | 4.833 | 6.063 | 6.083 | 0.929 | 4.527 | 5.662 | 5.685 | 0.950 | 3.736 | 4.709 | 4.725 | 0.957 |
| 29 | **3.653** | **4.641** | **4.662** | **0.958** | 4.825 | 6.036 | 6.059 | 0.929 | 4.518 | 5.651 | 5.669 | 0.950 | 3.722 | 4.692 | 4.707 | 0.957 |
| 30 | **3.651** | **4.655** | **4.681** | **0.959** | 4.822 | 6.029 | 6.056 | 0.929 | 4.524 | 5.654 | 5.673 | 0.950 | 3.745 | 4.729 | 4.751 | 0.956 |
| 31 | **3.665** | **4.674** | **4.701** | **0.959** | 4.835 | 6.065 | 6.092 | 0.929 | 4.507 | 5.635 | 5.652 | 0.950 | 3.741 | 4.723 | 4.740 | 0.956 |
| 32 | **3.622** | **4.631** | **4.651** | **0.960** | 4.816 | 6.036 | 6.054 | 0.929 | 4.493 | 5.622 | 5.642 | 0.950 | 3.735 | 4.695 | 4.713 | 0.957 |
| 33 | **3.641** | **4.642** | **4.669** | **0.960** | 4.831 | 6.040 | 6.063 | 0.929 | 4.511 | 5.625 | 5.652 | 0.950 | 3.720 | 4.697 | 4.713 | 0.957 |
| 34 | **3.638** | **4.637** | **4.656** | **0.960** | 4.870 | 6.089 | 6.113 | 0.928 | 4.516 | 5.642 | 5.657 | 0.950 | 3.745 | 4.729 | 4.742 | 0.956 |
| 35 | **3.621** | **4.618** | **4.647** | **0.960** | 4.836 | 6.053 | 6.075 | 0.929 | 4.524 | 5.646 | 5.665 | 0.950 | 3.731 | 4.712 | 4.735 | 0.957 |
| 36 | **3.628** | **4.638** | **4.662** | **0.960** | 4.839 | 6.053 | 6.071 | 0.929 | 4.506 | 5.617 | 5.634 | 0.950 | 3.706 | 4.684 | 4.701 | 0.957 |
| 37 | **3.635** | **4.614** | **4.638** | **0.959** | 4.842 | 6.054 | 6.074 | 0.929 | 4.510 | 5.628 | 5.648 | 0.950 | 3.745 | 4.754 | 4.768 | 0.956 |
| 38 | **3.626** | **4.634** | **4.660** | **0.960** | 4.841 | 6.066 | 6.091 | 0.928 | 4.533 | 5.666 | 5.684 | 0.949 | 3.738 | 4.726 | 4.743 | 0.956 |

## 4 Conclusion

The accurate prediction of body fat is important for assessing obesity and its related diseases. However, researchers find it challenging to analyse the large volumes of medical data generated. The main purpose of this study is to analyse and compare the prediction effectiveness of four well-known machine learning models (MLP, SVM, RF and XGBoost) when combined with three widely used feature extraction approaches (FA, PCA and ICA) for body fat prediction. The results presented in this paper are new in the context of body fat prediction; they could, therefore, provide a baseline for future research in this domain.

Experimental results showed that feature extraction methods can reduce features without incurring significant loss of information for body fat prediction. In Case 1, with only six extracted features, the prediction models exhibited better performance than the models without using feature extraction. This finding confirms the effectiveness of feature extraction. Among the comparison models, XGBoost with FA had the best approximation ability and high efficiency. With the increase in the number of extracted features, model performance can be further improved. For Case 2, PCA was the most effective in improving model

**Table 13. Experimental results for the MLP, SVM, RF, and XGBoost, based on the NHANES dataset, with ICA feature extraction (best results are highlighted in bold; # means the number of features).**

| # | MLP | | | | SVM | | | | RF | | | | XGBoost | | | |
|---|---|---|---|---|---|---|---|---|---|---|---|---|---|---|---|---|
| | MAC | SD | RMSE | MAC | MAC | SD | RMSE | MAC | MAC | SD | RMSE | MAC | MAC | SD | RMSE | MAC |
| 12 | 4.919 | 6.265 | 6.288 | 0.939 | 4.705 | 6.203 | 6.225 | 0.939 | 4.706 | 5.889 | 5.905 | 0.946 | **4.039** | **5.081** | **5.096** | **0.950** |
| 13 | 4.872 | 6.213 | 6.233 | 0.940 | 4.725 | 6.207 | 6.229 | 0.939 | 4.743 | 5.935 | 5.957 | 0.945 | **4.006** | **5.039** | **5.056** | **0.950** |
| 14 | 4.865 | 6.180 | 6.206 | 0.940 | 4.701 | 6.165 | 6.182 | 0.940 | 4.758 | 5.951 | 5.970 | 0.944 | **3.982** | **5.024** | **5.045** | **0.951** |
| 15 | 4.887 | 6.222 | 6.248 | 0.939 | 4.702 | 6.157 | 6.179 | 0.941 | 4.759 | 5.951 | 5.973 | 0.944 | **3.994** | **5.031** | **5.047** | **0.951** |
| 16 | 4.863 | 6.184 | 6.204 | 0.940 | 4.695 | 6.129 | 6.147 | 0.941 | 4.784 | 5.992 | 6.011 | 0.944 | **4.007** | **5.054** | **5.074** | **0.950** |
| 17 | 4.613 | 5.853 | 5.872 | 0.946 | 4.460 | 5.759 | 5.780 | 0.948 | 4.628 | 5.780 | 5.800 | 0.947 | **3.828** | **4.835** | **4.851** | **0.955** |
| 18 | 4.552 | 5.775 | 5.801 | 0.948 | 4.423 | 5.630 | 5.648 | 0.950 | 4.619 | 5.793 | 5.815 | 0.947 | **3.830** | **4.820** | **4.836** | **0.955** |
| 19 | 4.552 | 5.790 | 5.809 | 0.948 | 4.402 | 5.609 | 5.620 | 0.951 | 4.635 | 5.816 | 5.835 | 0.947 | **3.811** | **4.795** | **4.811** | **0.955** |
| 20 | 4.503 | 5.731 | 5.751 | 0.949 | 4.394 | 5.689 | 5.710 | 0.949 | 4.673 | 5.879 | 5.902 | 0.946 | **3.832** | **4.814** | **4.829** | **0.955** |
| 21 | 4.507 | 5.730 | 5.744 | 0.949 | 4.396 | 5.668 | 5.689 | 0.950 | 4.711 | 5.910 | 5.935 | 0.945 | **3.819** | **4.819** | **4.836** | **0.954** |
| 22 | 4.471 | 5.697 | 5.722 | 0.949 | 4.382 | 5.638 | 5.661 | 0.950 | 4.692 | 5.897 | 5.912 | 0.945 | **3.817** | **4.826** | **4.840** | **0.955** |
| 23 | 4.458 | 5.707 | 5.723 | 0.949 | 4.354 | 5.660 | 5.681 | 0.950 | 4.729 | 5.926 | 5.948 | 0.945 | **3.824** | **4.837** | **4.855** | **0.955** |
| 24 | 4.394 | 5.681 | 5.694 | 0.949 | 4.319 | 5.634 | 5.654 | 0.950 | 4.774 | 6.001 | 6.022 | 0.943 | **3.805** | **4.818** | **4.840** | **0.955** |
| 25 | 4.446 | 5.661 | 5.687 | 0.950 | 4.329 | 5.633 | 5.653 | 0.950 | 4.781 | 6.022 | 6.038 | 0.943 | **3.819** | **4.835** | **4.851** | **0.954** |
| 26 | 4.430 | 5.670 | 5.692 | 0.950 | 4.325 | 5.612 | 5.627 | 0.951 | 4.813 | 6.050 | 6.072 | 0.943 | **3.850** | **4.873** | **4.888** | **0.954** |
| 27 | 4.437 | 5.673 | 5.687 | 0.950 | 4.330 | 5.632 | 5.660 | 0.950 | 4.823 | 6.081 | 6.101 | 0.942 | **3.871** | **4.874** | **4.894** | **0.954** |
| 28 | 4.426 | 5.666 | 5.686 | 0.950 | 4.306 | 5.589 | 5.608 | 0.951 | 4.841 | 6.083 | 6.112 | 0.942 | **3.853** | **4.875** | **4.896** | **0.954** |
| 29 | 4.432 | 5.671 | 5.694 | 0.950 | 4.320 | 5.578 | 5.605 | 0.951 | 4.883 | 6.158 | 6.179 | 0.940 | **3.863** | **4.885** | **4.901** | **0.954** |
| 30 | 4.410 | 5.649 | 5.668 | 0.950 | 4.324 | 5.629 | 5.649 | 0.950 | 4.911 | 6.173 | 6.201 | 0.940 | **3.883** | **4.911** | **4.930** | **0.953** |
| 31 | 4.405 | 5.653 | 5.673 | 0.950 | 4.302 | 5.566 | 5.589 | 0.952 | 4.876 | 6.127 | 6.148 | 0.941 | **3.895** | **4.915** | **4.932** | **0.953** |
| 32 | 4.399 | 5.630 | 5.653 | 0.950 | 4.333 | 5.682 | 5.702 | 0.949 | 4.915 | 6.163 | 6.190 | 0.940 | **3.890** | **4.901** | **4.918** | **0.953** |
| 33 | 4.393 | 5.629 | 5.646 | 0.950 | 4.314 | 5.632 | 5.651 | 0.950 | 4.942 | 6.199 | 6.223 | 0.940 | **3.898** | **4.919** | **4.947** | **0.953** |
| 34 | 4.389 | 5.610 | 5.631 | 0.951 | 4.313 | 5.555 | 5.575 | 0.952 | 4.949 | 6.224 | 6.244 | 0.939 | **3.903** | **4.922** | **4.943** | **0.953** |
| 35 | 4.384 | 5.623 | 5.644 | 0.951 | 4.354 | 5.666 | 5.693 | 0.950 | 4.966 | 6.230 | 6.257 | 0.939 | **3.904** | **4.928** | **4.949** | **0.953** |
| 36 | 4.386 | 5.625 | 5.647 | 0.950 | 4.345 | 5.659 | 5.678 | 0.950 | 4.986 | 6.268 | 6.291 | 0.938 | **3.925** | **4.952** | **4.970** | **0.952** |
| 37 | 4.415 | 5.651 | 5.672 | 0.950 | 4.329 | 5.602 | 5.625 | 0.951 | 5.003 | 6.271 | 6.294 | 0.938 | **3.911** | **4.949** | **4.970** | **0.952** |
| 38 | 4.401 | 5.634 | 5.653 | 0.950 | 4.344 | 5.629 | 5.646 | 0.950 | 5.022 | 6.290 | 6.316 | 0.938 | **3.931** | **4.970** | **4.986** | **0.952** |

performance. Although the MLP with PCA had the best prediction accuracy, it required significantly more computation time. This means XGBoost is more appropriate for real-world applications, given its similar prediction accuracy and greater efficiency. Statistical analysis based on the Wilcoxon rank-sum test confirmed that feature extraction significantly improved the performance of MLP, SVM and XGBoost. This finding confirms the effectiveness of using feature extraction in these models. Although, the prediction models can be further improved slightly by increasing the number of extracted features, the number of features determined by the explained variance ratio was sufficient in both the considered cases.

The feature extraction results themselves are a novel contribution of this work. The results provided by XGBoost with PCA feature extraction could be used as the baseline for future research in related areas. In future studies, we plan to investigate ways to improve the feature extraction method specified for body fat datasets. Methods of improving the prediction model (e.g. an improved MLP [66]), using XGBoost with PCA as a baseline for body fat prediction, also need to be investigated. It is also worth noting that the findings of this work could be applied to other prediction problems with a large number of features, e.g., finance, engineering

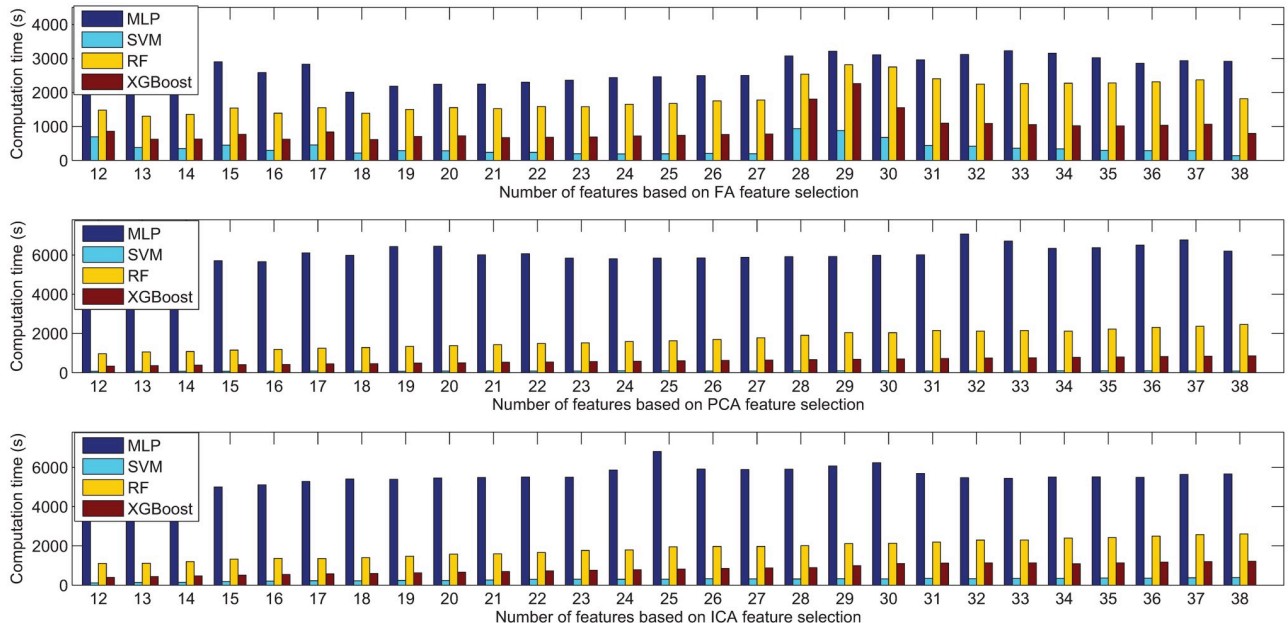

**Fig 12. Comparison results in terms of computation time based on FA, PCA, and ICA feature extraction for the NHANES dataset.**

and healthcare. Finally, we will explore other applications of analysing the body fat percentage. For example, applying domain knowledge to group body fat percentages into different disease classes in order to confirm the relationship between the body fat percentage and specific disease(s).

## Author Contributions

**Conceptualization:** Raymond Chiong.

**Formal analysis:** Zongwen Fan.

**Investigation:** Zongwen Fan, Raymond Chiong, Zhongyi Hu, Farshid Keivanian, Fabian Chiong.

**Methodology:** Zongwen Fan, Raymond Chiong.

**Software:** Zongwen Fan.

**Supervision:** Raymond Chiong, Zhongyi Hu, Fabian Chiong.

**Validation:** Zongwen Fan, Farshid Keivanian.

**Writing – original draft:** Zongwen Fan.

**Writing – review & editing:** Raymond Chiong, Zhongyi Hu, Fabian Chiong.

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
