## [Decision Letter · Decision Letter 0]

8 Oct 2021

PONE-D-21-28367Body fat prediction through feature extraction based on anthropometric and laboratory measurementsPLOS ONE

Dear Dr. Chiong,

Thank you for submitting your manuscript to PLOS ONE. After careful consideration, we feel that it has merit but does not fully meet PLOS ONE’s publication criteria as it currently stands. Therefore, we invite you to submit a revised version of the manuscript that addresses the points raised during the review process.

In particular:

The "methods" section is unclear and detailed setup of each model used during experiments is not presented,There is no deep analysis of feature extraction results,Presentation needs improvements:Fig. 7 and Fig. 9 are hard to read - please consider using log scale for vertical axis,Table 7 - please remove not needed zeroes from values,Selection of simple models is not discussed and justified. Models performing dynamic (contextual) data processing could give better results even for simple NNs in comparison with solutions with static FA/PCA/ICA (please see e.g. Sigma-if neural network),Conclusions are very shallow.Please submit your revised manuscript by Nov 22 2021 11:59PM. If you will need more time than this to complete your revisions, please reply to this message or contact the journal office at plosone@plos.org. Please include the following items when submitting your revised manuscript:A rebuttal letter that responds to each point raised by the academic editor and reviewer(s). You should upload this letter as a separate file labeled 'Response to Reviewers'.A marked-up copy of your manuscript that highlights changes made to the original version. You should upload this as a separate file labeled 'Revised Manuscript with Track Changes'.An unmarked version of your revised paper without tracked changes. You should upload this as a separate file labeled 'Manuscript'.

We look forward to receiving your revised manuscript.

Kind regards,

Maciej Huk, Ph.D.

Academic Editor

PLOS ONE

Journal Requirements:

https://journals.plos.org/plosone/s/file?id=ba62/PLOSOne_formatting_sample_title_authors_affiliations.pd

Reviewers' comments:

Reviewer's Responses to Questions

**Comments to the Author**

1. Is the manuscript technically sound, and do the data support the conclusions?

Reviewer #1: No

Reviewer #2: Yes

Reviewer #3: Yes

2. Has the statistical analysis been performed appropriately and rigorously? 

Reviewer #1: No

Reviewer #2: Yes

Reviewer #3: Yes

3. Have the authors made all data underlying the findings in their manuscript fully available?

Reviewer #1: Yes

Reviewer #2: Yes

Reviewer #3: Yes

4. Is the manuscript presented in an intelligible fashion and written in standard English?

Reviewer #1: Yes

Reviewer #2: Yes

Reviewer #3: Yes

5. Review Comments to the Author

Reviewer #1: In this paper, authors tried three standard feature extraction methods on two open datasets. Verified with three vaninal models.

The reason I reject this paper is that: this is a "toy paper". It will be a shame for this journal and machine learning community to publish something like this in 2021. I may grade a 'B' if this was a ML course project report from a junior university student.

- Motivation

What's the motivation of this paper? In the conclusion, the author states

"Experimental results showed that feature extraction methods can reduce features without incurring significant loss of information for body fat prediction."

"In Case 1, with only six extracted features, the prediction models exhibited better performance than the models without using feature extraction."

"For Case 2, PCA was the most effective in improving model performance. "

Do we need this paper to tell us this? Can't we find any similar sentences in any basic ML books for university students?

"Although the MLP with PCA had the best prediction accuracy, it required significantly more computation time. This means XGBoost is more appropriate for real-world applications, given its similar prediction accuracy and greater efficiency."

Didn't the author know that in real-world, complex neural networks have been used everywhere? This statement is so shabby and out of date.

- Experiment setup

Author didn't give detailed setup of each model.

"Case 1 contained 252 samples with 13 input features"

"After pre-processing, 862 samples with 39 features were obtained."

Is this a joke? With this amount of data and feature dimensions, authors are talking about "curse of dimension". Are we living in 1980s?

No deep analysis of feature extraction results.

- others

section 2.2 is waste of paper. You can find all those in wikipedia or any ML tutorial books/blogs/etc.

Reviewer #2: In this manuscript, the authors investigate the effectiveness of feature extraction for body fat prediction. Their results on two real body fat datasets demonstrate that the prediction models perform better on incorporating feature extraction for body fat prediction. The paper is well organized; however, I have some concerns:

1. The method section is poorly written and rather unclear on many points; You don't need to show every step/equation for well-known methods. Just summarize them and describe your settings (e.g., # of node, activation function, LR...)

2. The task is a continuous prediction (body fat percentage), and how did you perform SVM/RF on continuous prediction tasks? Did you group the body fat percentage into different classes?

3. Can authors compare them with methods without feature extraction (just min-max normalization on raw features)? This is an important baseline.

4. On page1 last row, what is "Lagrange multiplier measurements (features)"? Typo?

Reviewer #3: This work studies the problem of predicting body fat using a feature extraction method followed by a learning algorithm. For feature extraction, they tried ICA, PCA, and FA (factor analysis), and for learning and prediction, they used MLP, SVM, RF (random forest), and XGboost. According to the experimental validation work, XGBoost with FA has the best approximation ability. Overall the paper is easy to read and understand. Here are some issues I would like to see addressed:

1. State the novelty of the work.

2. Fix some notational inconsistencies, e.g., in section 2.2.1, stick with either e_i or error_i

3. Explain why you think ICA, PCA, and FA are the most useful feature extraction one should use and why MLP, SVM, RF, and XGBoost are picked as the prediction methods.

4. Why is this work confined only to a body fat prediction? Can it be used for any other prediction problem with only a large number of real-valued features?

6. PLOS authors have the option to publish the peer review history of their article (what does this mean?). If published, this will include your full peer review and any attached files.

Reviewer #1: No

Reviewer #2: No

Reviewer #3: No

---

## [Author Response · Author response to Decision Letter 0]

5 Nov 2021

Please refer to the response letter attached (appended to this document)

---

## [Decision Letter · Decision Letter 1]

18 Jan 2022

Body fat prediction through feature extraction based on anthropometric and laboratory measurements

PONE-D-21-28367R1

Dear Dr. Chiong,

We’re pleased to inform you that your manuscript has been judged scientifically suitable for publication and will be formally accepted for publication once it meets all outstanding technical requirements.

Kind regards,

Maciej Huk, Ph.D.

Academic Editor

PLOS ONE

Additional Editor Comments (optional):

Reviewers' comments:

Reviewer's Responses to Questions

**Comments to the Author**

1. If the authors have adequately addressed your comments raised in a previous round of review and you feel that this manuscript is now acceptable for publication, you may indicate that here to bypass the “Comments to the Author” section, enter your conflict of interest statement in the “Confidential to Editor” section, and submit your "Accept" recommendation.

Reviewer #2: All comments have been addressed

Reviewer #3: All comments have been addressed

2. Is the manuscript technically sound, and do the data support the conclusions?

Reviewer #2: Yes

Reviewer #3: Yes

3. Has the statistical analysis been performed appropriately and rigorously? 

Reviewer #2: (No Response)

Reviewer #3: Yes

4. Have the authors made all data underlying the findings in their manuscript fully available?

Reviewer #2: Yes

Reviewer #3: Yes

5. Is the manuscript presented in an intelligible fashion and written in standard English?

Reviewer #2: Yes

Reviewer #3: Yes

6. Review Comments to the Author

Reviewer #2: (No Response)

Reviewer #3: All of my comments have been adequately addressed. Thank you. The revised manuscript looks much better now.

7. PLOS authors have the option to publish the peer review history of their article (what does this mean?). If published, this will include your full peer review and any attached files.

Reviewer #2: No

Reviewer #3: No

---

## [Editor Report · Acceptance letter]

28 Jan 2022

PONE-D-21-28367R1 

Body fat prediction through feature extraction based on anthropometric and laboratory measurements" 

Dear Dr. Chiong:

I'm pleased to inform you that your manuscript has been deemed suitable for publication in PLOS ONE. Congratulations! Your manuscript is now with our production department. 

Kind regards, 

on behalf of

Dr. Maciej Huk 

Academic Editor

PLOS ONE